# Growth Curves of Chinese Children with Androgen Insensitivity Syndrome: A Multicenter Registry Study

**DOI:** 10.3390/jpm12050771

**Published:** 2022-05-10

**Authors:** Xiu Zhao, Zhe Su, Shaoke Chen, Xiumin Wang, Yu Yang, Linqi Chen, Li Liang, Geli Liu, Yi Wang, Yanning Song, Lijun Fan, Xiaoya Ren, Chunxiu Gong

**Affiliations:** 1Department of Endocrinology, Shenzhen Children’s Hospital, Shenzhen 518028, China; zoeman221@163.com (X.Z.); su_zhe@126.com (Z.S.); 2Department of Pediatrics, Second Affiliated Hospital of Guangxi Medical University, Nanning 530007, China; chenshaoke123@163.com; 3Department of Endocrinology, Shanghai Children’s Medical Center, Shanghai Jiaotong University, Shanghai 200120, China; wangxiumin@scmc.com.cn; 4Department of Endocrinology, Jiangxi Provincial Children’s Hospital, Nanchang 330006, China; yan-yu5168@126.com; 5Department of Endocrinology, Children’s Hospital of Soochow University, Suzhou 215008, China; clq631203@aliyun.com; 6Department of Pediatrics, First Affiliated Hospital of Zhejiang University, Hangzhou 310006, China; zdliangli@163.com; 7Department of Pediatrics, Tianjin Medical University General Hospital, Tianjin 300052, China; zyy@163.com; 8Center of Endocrinology, Genetics and Metabolism, Beijing Children’s Hospital, Capital Medical University, National Center for Children’s Health, Beijing 100045, China; wangyi880512@163.com (Y.W.); songyanning123@163.com (Y.S.); fanlijun0916@163.com (L.F.); renxya2014@163.com (X.R.)

**Keywords:** androgen insensitivity syndrome, growth, growth curves, androgen receptor, androgens

## Abstract

Objective: To provide new information about androgen insensitivity syndrome (AIS), we studied growth patterns in Chinese children with AIS. Subjects: Data are from 118 untreated AIS patients who were admitted to eight pediatric endocrine centers from January 2010 to December 2019. Methods: In this retrospective cohort study, clinical data were collected from a multicenter database. We compared physical assessment data among AIS patients and standard growth charts for Chinese pediatric population. Results: 1. Children with AIS grew slightly less than the mean before 6 months of age, and then, height gradually increased before 12 years of age, from the median to +1 standard deviation (SD), according to the standard reference for Chinese pediatric population. After 12 years of age, height showed differently in profiles: The mean height in AIS patients gradually decreased from the mean to −1 SD, according to the standard for Chinese boys, and increased from the mean to +2 SD, according to the standard for Chinese girls. 2. The weights of children with AIS were greater than the mean standards of Chinese pediatric population from newborn to 11 years of age. From 12–16 years of age, the mean weight of children with AIS showed different profiles, from the mean to −1 SD, according to the standard for Chinese boys and from the mean to +1.5 SD, according to the standard for Chinese girls. 3. Weight standard deviation (WtSDS) and target height (THt) in northern Chinese AIS patients were significantly higher than those from the southern region (*p* = 0.035, 0.005, respectively). Age in northern Chinese AIS patients was significantly younger than those from the southern region (*p* = 0.034). No difference was found among birth weight (BW), birth length (BL), height standard deviation (HtSDS) and body mass index (BMI) in AIS patients from different regions (*p* > 0.05). 4. HtSDS and WtSDS in complete AIS (CAIS) patients were higher than those in partial AIS (PAIS) patients without significant difference (*p* > 0.05). Conclusions: Growth of children with AIS varied to different degrees. AIS patients seemed not to experience a puberty growth spurt. CAIS and PAIS patients show little difference in their growth. Regional differences have no effect on the height but influence the weight of AIS patients.

## 1. Introduction

Androgen insensitivity syndrome (AIS, OMIM 300068) is the most common 46, XY disorder of sexual development (DSD) in western countries [1] and the second most common in China. AIS is an X-linked inherited disorder due to a mutation in the *AR* gene (OMIM 313700) [2,3,4,5,6]. The endocrine profile is characterized by elevated or normal basal serum testosterone levels associated with high serum luteinizing hormone (LH) levels in AIS [6,7]. The clinical manifestations of AIS present a broad spectrum of phenotypes, including normal female external genitalia, ambiguous genitalia, cryptorchidism, micropenises, hypospadias and even a normal male phenotype [4,8,9,10,11,12]. According to the phenotype and residual receptor activity, AIS can be divided as three types: complete AIS (CAIS), partial AIS (PAIS) and a mild form of AIS (MAIS). There are different cell types and mechanisms that mediate the effects of sex hormones on bone [13]. Androgens have a direct effect on androgen receptors (AR) and an indirect effect on estrogen receptors α (ERα) via aromatization [14]. Moreover, androgens can stimulate the secretion of growth hormones (GH) and also have an indirect influence on circulating insulin-like growth factor 1 (IGF-1) via peripheral and central aromatization [15,16]. Therefore, children with AIS may show different growth patterns from healthy children due to the influence of many mechanisms. However, previous studies often focused on their mental health and gender distribution rather than the growth patterns. We currently only have a preliminary understanding of the growth pattern of AIS children in a few studies, which suggested that the final adult height (FAH) in patients with CAIS often lagged behind males [3,17].

In our previous multicenter cohort study, we found that 141 children with 5 alpha-reductase type 2 deficiency (5αRD), which has an impaired androgen synthesis, had a unique growth curve [18]. In this research, we studied the growth patterns of AIS patients and found differences between AIS patients and Chinese pediatric population.

## 2. Materials and Methods

### 2.1. Patients and Controls

From January 2010 to December 2019, 146 Chinese patients with untreated AIS aged 0.08–16.00 years (median age: 2.17 years) from eight pediatric endocrine centers in China were enrolled in this retrospective cohort study. The inclusion criteria were as follows: (1) patients whose clinical manifestations included ambiguous genitalia, labial mass or inguinal hernia with normal female-type external genitalia, hypospadias, micropenises and others; (2) patients who were diagnosed with AIS according to their clinical manifestations, hormone tests and identified pathogenic or like pathogenic *AR* gene variants; (3) patients with the karyotype 46, XY; (4) patients for whom growth data could be obtained before sex hormone treatment or gonadectomy; (5) patients were born at full-term gestation.; and (6) patients whose parents or caregivers provided informed consent for genetic analysis. The exclusion criteria were as follows: (1) patients with abnormal functioning of the liver or kidney, malformations, or other systemic diseases that may have affected physical development; (2) patients with another type of 46, XY DSD as diagnosed by biochemical diagnosis and genetic confirmation. According to the inclusion and exclusion criteria, 118 patients were enrolled, and 28 patients were excluded from the study. This study was approved by the ethics committee of Beijing Children’s Hospital. The general growth reference values of Chinese pediatric population were used as standard controls. The height and weight standardized growth charts for Chinese children aged 0 to 18 years have been published in the literature [19].

### 2.2. Methods

#### 2.2.1. Data Collection

The collected anthropometric data were as follows: height (Ht), weight (Wt), birth length (BL), birth weight (BWt), family history, external genitalia, and puberty development. Puberty was described by the Tanner stage [20]. Height and weight were measured using standardized equipment. Height was measured as the orthostatic height without shoes in patients aged >3 years old and the supine length in patients aged <3 years old using an infantometer. For each case, height and weight were measured three times by experienced nurses, and the average value was taken. Experienced endocrinologists performed the examination of the external genitalia and determined the Tanner stage. Height and weight values were approximated to the nearest 0.1 cm and 0.1 kg, respectively. Body mass index (BMI), target height (THt), height standard deviation score (HtSDS) and weight standard deviation score (WtSDS) were calculated. The calculator formulae were as follow: BMI = Wt (kg) ÷ Ht (m)^2^, THt = (father’s Ht + mother’s Ht ± 13) ÷ 2 (cm).

#### 2.2.2. Gene Analysis

All 118 patients underwent whole exon sequencing (WES) and received a molecular diagnosis of AIS. Five to ten milliliters of peripheral blood was collected in disposable vacuum tubes for genetic testing. Genomic DNA was isolated using the QIAamp DNA Blood Mini Kit (Qiagen, Hilden, Germany) according to the manufacturer’s instructions. After construction according to the standard protocol, whole exon sequencing (100×) of the libraries was performed with the SureSelect Human All Exon V6 array on the Illumina HiSeq X Ten Platform with the PE150 strategy. A standard bioinformatics pipeline was utilized for variant identification with the help of Genome Analysis Toolkit (GATK, Cambridge, America) [21] software following the best practice guidelines recommended by the GATK [22,23]. Candidate variants were retained as follows: [1] rare variants with a minor allele frequency of <1% in the ExAC, dbSNP, 1000 Genomes, gnomAD and local databases, and [2] functional variants including frameshift, splice, nonsense, missense and synonymous variants that can affect splicing. Then, a hypothesis-free approach is used to analyze all phenotype-related genes. Sequence variants were confirmed by Sanger sequencing. The eight exons of the *AR* gene were covered. The pathogenic *AR* gene variants were confirmed in individuals carrying hemizygotic variants inherited from mothers or de novo. Nucleotide sequences of the *AR* genes were compared with published data. The pathogenicity of unreported *AR* variants was tested, and pathogenic analysis was performed using two software packages, ‘Polyphen’ (http://genetics.bwh.harvard.edu/pph2/, accessed on 1 January 2020) and ‘SIFT’ (https://sift.bii.a-star.edu.sg/www/code.html, accessed on 1 January 2020), and two webtools, ‘Mutation Taster’ (http://www.mutationtaster.org, accessed on 1 January 2020) and Mutation Assessor (http://mutationassessor.org/r3/, accessed on 1 January 2020). The interpretation of gene pathogenicity was based on the American College of Medical Genetics (ACMG, America) [24]. The researchers reviewed all gene results.

#### 2.2.3. Growth Curve Plotting and Data Analysis

The internationally accepted method (λ-median-coefficient of variation, LMS) generated standard curves [25]. The calculations for the growth curves were performed using LMS-chartmaker Pro software, and curves were drawn using the GraphPad Prism 6 software. Growth curves of children with AIS were generated for height for age, weight for age and BMI for age. Growth curves (P3, P10, P25, P50, P75, P90 and P97 percentile curves as well as −2 SD, −1 SD, 0 SD, +1 SD, and +2 SD standard deviation curves) for children aged 0–36 months and 3–16 years were constructed. SPSS 23.0 software was used for statistical analyses. Data pertaining to quantitative variables are expressed as the mean ± standard deviations (SDs) or quartiles. Intergroup differences were assessed using the Student’s *t*-test for normally distributed data and the Mann–Whitney U test for nonnormally distributed data. *p* < 0.05 was considered statistically significant.

## 3. Results

### 3.1. General Data of Chinese AIS Patients

In the study, 118 patients with AIS (age: 2.17 (0.94, 6.92) years) were all Chinese children. There were 67 patients with AIS aged <3 years and 51 patients aged 3–16 years. Forty-two and 76 patients were diagnosed as CAIS and PAIS, respectively. No patient was diagnosed as MAIS. Among them, 73 children with AIS were raised as female, including 31 children with PAIS and 42 children with CAIS. Those children raised as male were all PAIS patients (*n* = 45). In PAIS patients, 59.21% and 40.79% were raised as male and female, respectively. The clinical manifestations of all patients are shown in Figure 1. The manifestations of CAIS were labial mass or inguinal hernia with normal female-type external genitalia. The clinical manifestations of PAIS were hypospadias with micropenises or cryptorchidism, ambiguous genitalia, isolated micropenises and isolated hypospadias in turn.

The molecular diagnosis in all AIS patients was proved to have pathogenic or like pathogenic variants in *AR* according to the rules of ACMG. Three out of 118 AIS patients are de novo *AR* variants in the study.

### 3.2. Height-for-Age Growth Curve

The height-for-age growth curves of children with AIS are shown in Figure 2 and Figure 3, which represent the height growth charts for AIS patients compared to the height standard for Chinese pediatric population. The BL of children with AIS was within the normal range of Chinese newborns. Length values in AIS patients aged < 6 months old were between the mean and −0.3 SD of the standard values of Chinese boys and girls. Then before the age of 11 years, height in children with AIS increased compared with Chinese boys and girls and the mean height values ranged between the mean and +1 SD of Chinese boys and girls. Beyond the age of 12 years, the mean height of children with AIS was lower than that of Chinese boys and ranged from the mean to −1 SD of Chinese boys, whereas the mean height of children with AIS was higher than that of Chinese girls, ranging from the mean to +2 SD of Chinese girls. Height of AIS children did not rise sharply with the age compared with those of the Chinese girls and boys during the age of 8–16 years. In conclusion, the growth patterns of height in children with AIS deviated somewhat from those in Chinese pediatric population.

### 3.3. Weight-for-Age Growth Curve

The weight-for-age growth curves of children with AIS are shown in Figure 4. Figure 3 presents the weight growth charts for AIS patients compared to the weight standard for Chinese pediatric population. The BWt of children with AIS was within the normal range of Chinese newborns. Comparison to the Chinese boys’ growth chart, the mean weight in children with AIS was higher and ranged from the mean to +1 SD of Chinese boys until 11 years of age. Beyond the age of 12 years, the mean weight of children with AIS was lower than that of the Chinese boys, ranging from the mean to −1 SD of Chinese boys. Compared to the Chinese girls’ growth chart, the mean weight in AIS patients aged <16 years was higher and ranged the mean to +1.5 SD of Chinese girls. In conclusion, the weights of children with AIS were higher than that of Chinese girls aged <16 years and Chinese boys aged <12 years.

### 3.4. BMI-for-Age Growth Curve

The BMI-for-age growth curves of children with AIS are shown in Figure 5. Figure 3 represents the BMI growth charts for AIS patients compared to the standard BMI for Chinese pediatric population. Compared to the Chinese boys’ growth chart, the mean BMI value in children with AIS was higher and ranged from the mean to +1.3 SD of Chinese boys until the age of 10 years. The mean BMI of children with AIS was similar to that of Chinese boys during the age of 11–16 years. Compared to the Chinese girls’ growth chart, the mean BMI in AIS patients aged <11 years was higher and ranged from the mean to +2 SD of Chinese girls. Beyond the age of 12 years, the BMI of children with AIS decreased and the mean values of BMI ranged from the mean to +0.5 SD of Chinese girls. In conclusion, the BMIs in children with AIS were higher than those in Chinese boys aged <11 years and girls aged <16 years.

### 3.5. Regional Differences in Chinese AIS Patients

All 118 AIS patients came from 25 provinces and municipalities in China. According to the geographical boundaries of the Qinling Mountains and Huai River, China can be divided into southern and northern regions. Geographic difference analysis was based on northern and southern regions. WtSDS and THt in AIS patients from northern China were significantly higher than those from the southern region (*t* = −2.426, *Z* = −2.804, *p* = 0.017, 0.005, respectively). The age of AIS patients from northern China was significantly younger than that from the southern region (*Z* = −2.123, *p* = 0.034). No significant difference was found in BW, BL, HtSDS and BMI in AIS patients from the northern and southern regions. (See Table 1)

### 3.6. Physical Assessment in CAIS and PAIS Patients

There were 76 PAIS patients and 42 CAIS patients in the study. BWt, BL and THt were similar in PAIS and CAIS patients. HtSDS and WtSDS in PAIS patients were higher than those of CAIS patients without significant difference (*t* = −0.902, −0.171, *p* = 0.369, 0.864, respectively). HtSDS and WtSDS in PAIS patients aged 3–16 years and aged 0-36 months showed no significant difference to those of CAIS patients (*t* = −1.206, −0.204, −0.561, 0.278, *p* = 0.234, 0.839, 0.578, 0.782, respectively). THt in PAIS patients aged 0–36 months showed a significant difference to that of CAIS patients (*Z* = 2.558, *p* = 0.011). (See Table 2).

## 4. Discussion and Conclusions

AIS is a common type of 46, XY DSD. Approximately 20% of PAIS and 100% of CAIS were raised as female [26]. In our study, 59.21% of PAIS and 100% of CAIS were raised as female. The phenotype of PAIS varied very widely. The most common manifestation of PAIS (86%) in this study showed hypospadias with micropenises or cryptorchidism and ambiguous genitalia. The phenotype of CAIS was labial mass or inguinal hernia with normal female-type external genitalia. *AR* variants in most AIS patients are inherited from the patients’ mother in the study. Only three AIS patients are de novo *AR* variants, which might be due to a selection bias.

With advances in genetic testing technologies, an increasing number of AIS patients are being diagnosed, even during the newborn period. Growth is a special manifestation in children. Therefore, it is necessary to study the growth pattern of children with AIS. A few studies focus on the final adult height (FAH) of AIS patients. FAH in CAIS was lower than in adult males [3,17]. FAH was lower than the male average height in 38.10% of the 21 PAIS patients and higher than the male average height in the remaining 61.90%, with a higher percentage of reduced FAH in EMS < 5 [27]. In our study, the FAHs of patient with CAIS and PAIS were 167.0 ± 1.1 cm and 171.8 ± 0.8 cm, respectively, which are lower than the median height of adult males and higher than the median height of adult females, just as described in the above reports. The different residual activity of AR may explain the reason for the differences in growth in PAIS and CAIS patients. Additionally, T is an important factor for determining body composition, especially abdominal obesity and lean weight in males. AR plays important roles in male metabolism by affecting the energy balance, adiposity and insulin sensitivity via change of dynamic and oxygen consuming, transcripts for the thermogenetic uncoupling protein 1, adiponectin and lipolysis [28]. In a long-term study, in 18 patients with CAIS (14 after gonadectomy and 4 with intact testes), the prevalence of obesity was higher (16.7% vs. 3.6%) than in those with a normal BMI [29]. In our study, the BMIs and weight in children with AIS were higher than those in Chinese boys aged <11 years and Chinese girls aged <16 years. These show that gonadal hormones and AR play indispensable roles in the distribution of lean mass and adipose tissue, as well as body composition as a whole.

This study was the first to show the growth curves of children with AIS. Our research showed that the BL and BWt in newborns with AIS was within the normal range of Chinese newborns, similar to the results of previous reports [30]. Compared to the standard reference of Chinese pediatric population, the height of children with AIS was slightly less before the age of 6 months, and then the speed of growth gradually increased before the age of 11 years, ranging from the mean to +1 SD. Beyond 12 years of age, the height of AIS patients decreased gradually, ranging from the mean to −1 SD compared to the mean standard for Chinese boys. While compared to the mean standard for Chinese girls, the mean height of AIS patients increased and ranged from the mean to +2 SD. Compared to the growth curve of AIS patients and Chinese pediatric population, height of AIS patients showed a steady growth rate after the age of 8 years old and lacked the puberty growth spurt that the Chinese pediatric population should experience. Sex hormones play a very important role in children’s growth through many pathways. Increased levels of androgens are present in individuals with AIS. Despite the defect of AR, elevated androgens can exert their growth-regulating effects through other pathways, such as peripheral and central aromatization. In addition, the growth-controlling region on the human Y chromosome also has an effect on growth [31]. In previous reports, LH and follicle-stimulating hormone (FSH) levels in AIS patients increased [7,32,33,34,35]. The increased level of LH and insufficient feedback in the hypothalamic-pituitary-gonadal (HPG) axis make the onset time of puberty earlier than usual in AIS patients [32,34,36,37]. As a result, the initiation age of puberty in AIS patients is close to the initiation age in females and earlier than the initiation age in males, and the age of attaining adult height in AIS is between that age in males and females [32,38,39]. At the same time, the timing of epiphyseal maturation corresponds more closely to the male pattern than to the female pattern [39]. All these factors may have an effect on the growth pattern and the FAH of AIS patients.

China is a continental country. Regional differences could be relevant and may play a role in growth. According to the geographical boundaries of the Qinling Mountains and Huai River, China can be divided into southern and northern regions. The height and weight in northern people are greater than those in southern people [40,41]. We compared the physical assessment in Chinses AIS patients from the northern and southern regions to estimate the regional differences in our multicenter study. WtSDS and THt in AIS patients from northern China were significantly higher than those from the southern region. The age of AIS patients from northern China were significantly younger than those from the southern region. No significant differences were found in BW, BL, HtSDS and BMI in southern and northern AIS patients. Thus, the weight of AIS patients shows a regional difference and may have an influence on the weight curve, whereas height shows no regional difference. In addition to geographical factor, the difference in WtSDS of AIS patients may be due to other factors such as diet, exercise, socio-economic factors, gonadal hormones and parental education and occupation [42,43,44].

## 5. Study Limitations

Due to the limitation of sample size, growth curves could not be constructed according to CAIS and PAIS. Therefore, we will continue our studies and obtain more data for AIS patients. In the study, the virilization score (Sinnercker score or external genitalia masculinization score) was not measured because some patients’ medical records were not well documented. In conclusion, this study found that the growth of children with AIS deviated from that of Chinese boys or girls to different degrees. AIS patients did not experience a puberty growth spurt. CAIS and PAIS patients show some difference in growth. While regional differences had no effect on the height of AIS patients, there was an effect on the weight of AIS patients.

## Figures and Tables

**Figure 1 jpm-12-00771-f001:**
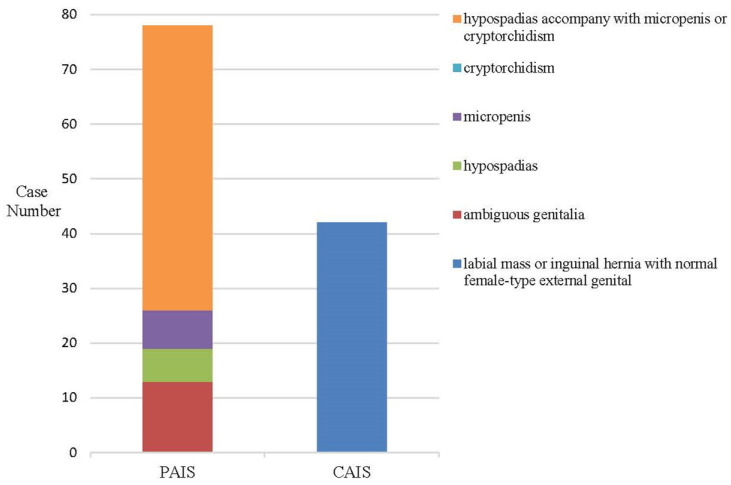
Phenotype of 118 Chinese children with AIS.

**Figure 2 jpm-12-00771-f002:**
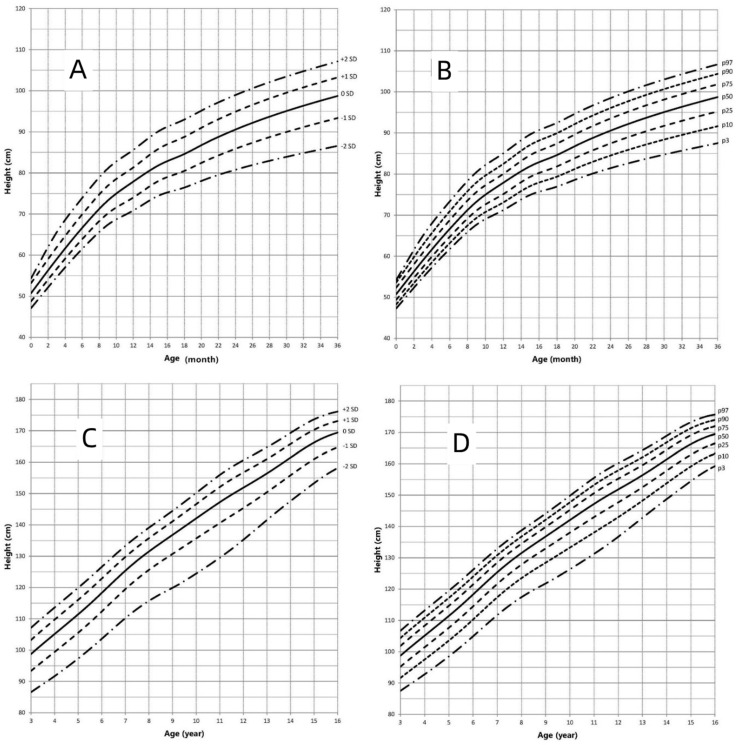
Height in children with AIS (0–16 years old). (**A**): Standard deviation curve (0–36 months old); (**B**): Percentile curve (0–36 months old); (**C**): Standard deviation curve (3–16 years old); (**D**): Percentile curve (3–16 years old).

**Figure 3 jpm-12-00771-f003:**
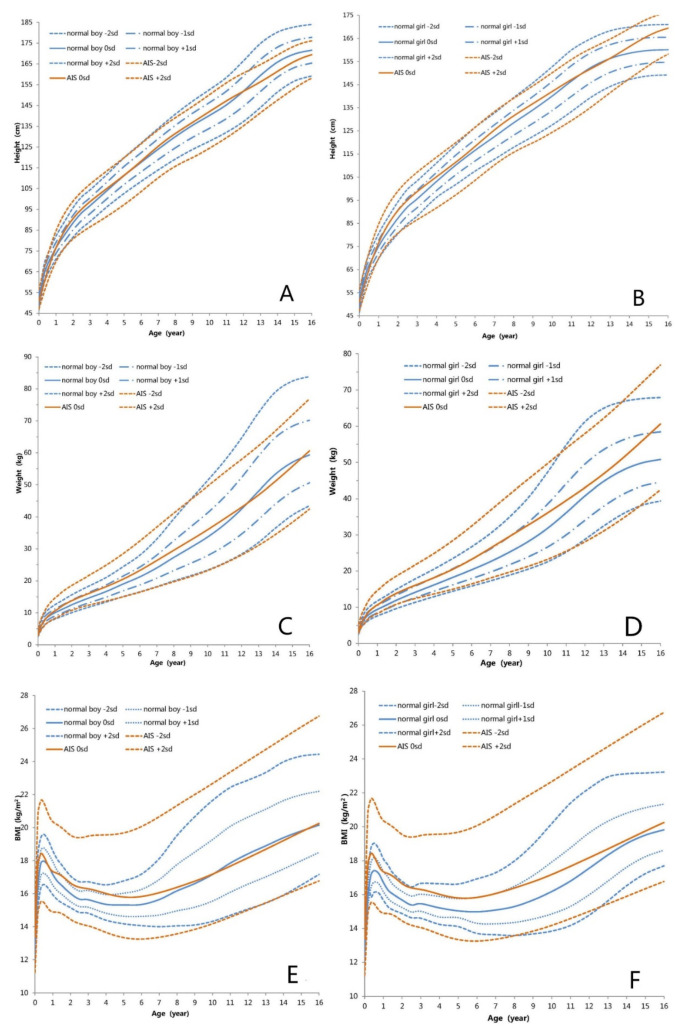
Standard deviation height, weight and BMI growth curves between children with AIS and Chinese boys and girls (0–16 years old). (**A**): Height standard deviation curve of Chinese boys; (**B**): Height standard deviation curve of Chinese girls; (**C**): Weight standard deviation curve of Chinese boys; (**D**): Weight standard deviation curve of Chinese girls; (**E**): BMI standard deviation curve of Chinese boys; (**F**): BMI standard deviation curve of Chinese girls; Blue indicates normal Chinese boys or girls. Red indicates patients with AIS. sd = standard deviation scores.

**Figure 4 jpm-12-00771-f004:**
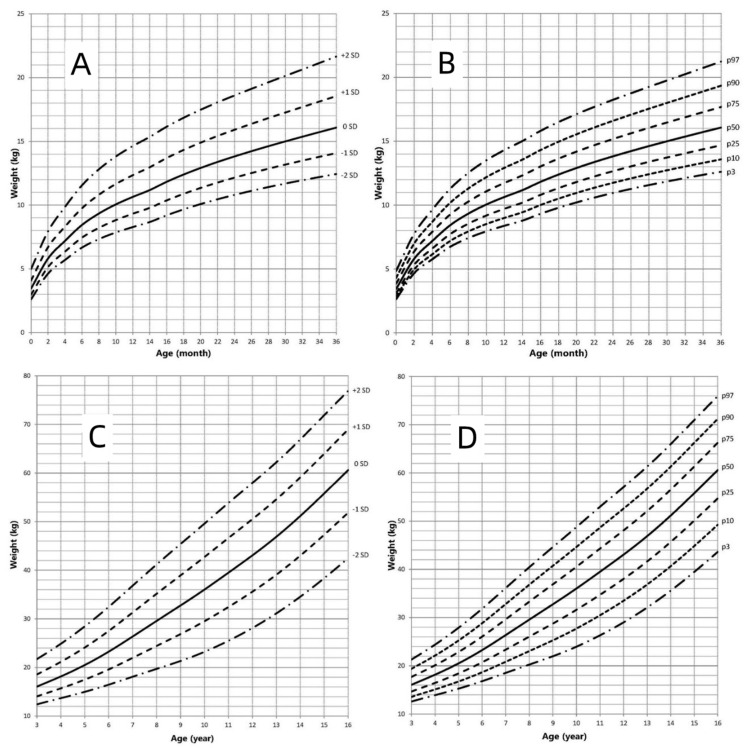
Weight in children with AIS (0–16 years old). (**A**): Standard deviation curve (0–36 months old); (**B**): Percentile curve (0–36 months old); (**C**): Standard deviation curve (3–16 years old); (**D**): Percentile curve (3–16 years old).

**Figure 5 jpm-12-00771-f005:**
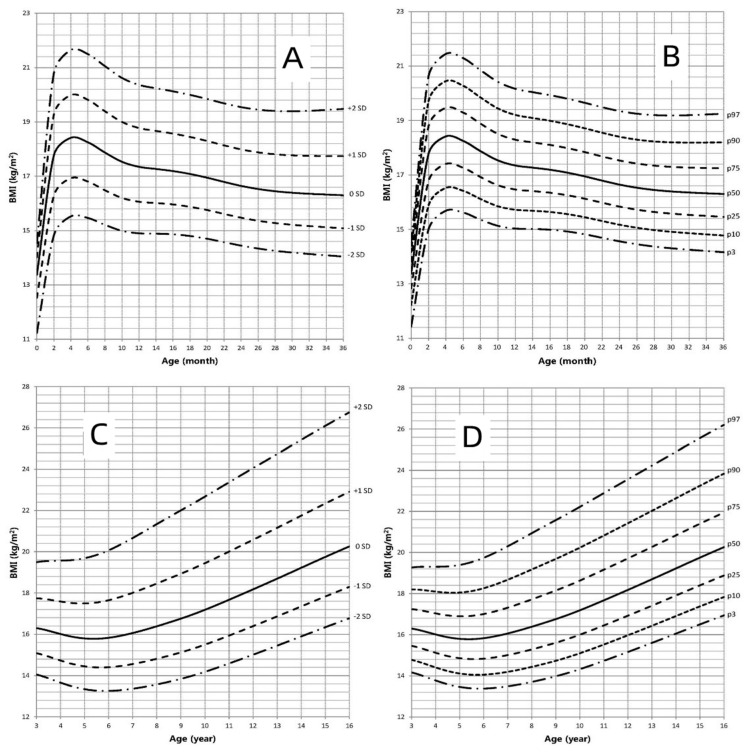
BMI in children with AIS (0–16 years old). (**A**): Standard deviation curve (0–36 months old); (**B**): Percentile curve (0–36 months old); (**C**): Standard deviation curve (3–16 years old); (**D**): Percentile curve (3–16 years old).

**Table 1 jpm-12-00771-t001:** Clinical parameters of children with AIS in the northern and southern regions of China.

Region	*n*	Age (Year)	THt (cm)	BWt (kg)	BL (cm)	HtSDS	WtSDS	BMI (kg/m^2^)
South	39	4.08 (1.16, 8.98)	169.64 ± 3.70 ^*^	3.40 (3.15, 3.50)	50 (49, 51)	0.25 ± 1.36	0.24 ± 1.21	17.45 ± 2.92
North	79	1.85 (0.92, 5.08) *	173.00 (170.00, 174.63)	3.40 (3.20, 3.82)	50 (50, 50)	0.64 ± 1.34	0.82 ± 1.23 *	17.21 (16.33, 18.90)
Total	118	2.17 (1.00, 6.19)	172.50 (171.00, 176.00)	3.40 (3.20, 3.76)	50 (50, 50)	0.51 ± 1.35	0.62 ± 1.25	17.28 (16.30, 18.93)
*t/Z*		−2.123	−2.804	−0.247	−0.144	−1.499	−2.426	−0.781
*p*		0.034	0.005	0.805	0.885	0.137	0.017	0.436

*p* < 0.05, * Southern region vs. northern region using Student’s *t*-test for normally distributed data and Mann–Whitney U test for non-normally distributed data. THt = target height standard deviation score, BWt = birth weight, BL = birth length, HtSDS = height standard deviation score, WtSDS = weight standard deviation score, BMI = body mass index.

**Table 2 jpm-12-00771-t002:** Clinical parameters of Chinese children with PAIS and CAIS.

Groups		*n*	Age (Year)	THt (cm)	BWt (kg)	BL (cm)	HtSDS	WtSDS	BMI (kg/m^2^)
PAIS	0–36 months	42	1.08 ± 0.72	173.50 (171.50, 176.00)	3.46 ± 0.51	50.00 (50.00, 50.00)	0.93 ± 1.45	0.30 ± 1.17	17.16 (16.33, 18.95)
	3–16 years	34	7.13 (4.94, 11.94)	171.00 ± 3.46	3.40 ± 0.52	50.00 (48.25, 50.25)	0.60 ± 1.12	0.80 ± 1.22	17.57 (16.43, 20.28)
	Total	76	2.25 (1.00, 6.56)	173.00 (170.00, 174.63)	3.45 ± 0.51	50.00 (49.00, 51.00)	0.66 ± 1.32	0.65 ± 1.18	17.37 (16.35, 19.35)
CAIS	0–36 months	25	1.00 ± 0.74	176.00 ± 3.54 *	3.30 (3.20, 3.50)	50.00 (50.00, 51.00)	0.72 ± 1.49	0.59 ± 1.29	17.00 (16.08, 17.99)
	3–16 years	17	7.42 (4.37, 8.75)	173.00 (172.25, 176.50)	3.40 (3.15, 3.80)	50.00 (50.00, 51.75)	−0.1 ± 1.16	0.65 ± 1.31	17.29 (15.02, 19.21)
	Total	42	2.10 (0.93, 6.20)	174.75 (169.90, 175.13)	3.35 (3.20, 3.50)	50.00 (50.00, 51.00)	0.43 ± 1.37	0.61 ± 1.29	17.15 (15.88, 18.42)
	0–36 months		0.487	2.558	0.709	0.927	−0.204	0.278	-0.318
*t/Z*	3–16 years		−1.099	0.107	0.577	−0.209	−1.206	−0.561	−0.909
	Total		−0.481	−1.040	−0.769	−0.901	0.902	0.171	−0.925
*p*	0–36 months		0.627	0.011	0.479	0.955	0.839	0.782	0.251
	3–16 years		0.272	0.109	0.584	0.263	0.234	0.578	0.363
	Total		0.631	0.298	0.442	0.368	0.369	0.864	0.355

*p* < 0.05, * Complete androgen insensitivity (CAIS) vs. partial androgen insensitivity (PAIS) using Student’s *t*-test for normally distributed data and Mann–Whitney U test for non-normally distributed data. CAIS = complete androgen insensitivity, THt = target height standard deviation score, BWt = birth weight, BL = birth length, HtSDS = height standard deviation score, WtSDS = weight standard deviation score, BMI = body mass index.

## Data Availability

The dataset analyzed in the current study is available from the corresponding author upon reasonable request.

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
