# Peer review of "Growth Curves of Chinese Children with Androgen Insensitivity Syndrome: A Multicenter Registry Study"

_jpm, 2022, doi:10.3390/jpm12050771_

Round 1
Reviewer 1 Report
The authors describe growth patterns in AIS children/adolescents compared to control children/adolescents in China. The presented data are interesting but the manuscript needs some revision.
- Generally, the spelling and wording should be revised:
- Abbreviations should be used only after the term was described
- The sentence in the abstract is not correct: WtSDS, target height (THt) and age in northern Chinese AIS patients were significantly higher and younger than those from the southern region. Either the authors say that it was significantly different or they split the sentence in two.
- In the methods section: do the authors mean born at gestational age <38 weeks instead of patients with a gestational age <38 weeks?
- The term genitalia should be used throughout the manuscript (not genitals).
- I would suggest the authors not to compare AIS with normal children as it implies AIS as an abnormal person. Rather use the term unaffected control children.
- The scientific purpose of the study is not clear. The rational on why this study was performed has to be elaborated in the introduction.
- The authors state a significant difference in WTSDS and Tht between the northern and southern population. The author should correct the data of these two parameters for this difference.
- The authors mention novel AR-mutations. They should not be called mutations, but variants unless the authors show that these new variants are mutations.
Author Response
1.Generally, the spelling and wording should be revised:
1) Abbreviations should be used only after the term was described
Author reply:Thank you for your recommend. I have revised the Abbreviations (page2, line16, page2, line25, page3, line1, page3, line3-4).
2) The sentence in the abstract is not correct: WtSDS, target height (THt) and age in northern Chinese AIS patients were significantly higher and younger than those from the southern region. Either the authors say that it was significantly different or they split the sentence in two.
Author reply:Thank you for your recommend. I have revised the sentence according to your suggestion (page2, line25-page3, line1).
3) In the methods section: do the authors mean born at gestational age <38 weeks instead of patients with a gestational age <38 weeks?
Author reply:Thank you for your recommend. I have revised the sentence ‘patients were born at the full-term gestation’ (page5, line4).
4) The term genitalia should be used throughout the manuscript (not genitals).
Author reply:Thank you for your recommend. I have changed the term genitals to the term genitalia (page3, line29, page7, line10).
5) I would suggest the authors not to compare AIS with normal children as it implies AIS as an abnormal person. Rather use the term unaffected control children.
Author reply:Thank you for your recommend. The other reviewer gave me the same comment. He suggested to change the term normal children to the term Chinese pediatric population. I have revised the term normal children. (page2, line13, page2, line17, page2, line22, page4, line8, page4, line16, page5, line7, page8, line3-4, page8, line14, page11, line7-8, page12, line7-8, page17, line7, page17, line12-13, page17, line14)
6) The scientific purpose of the study is not clear. The rational on why this study was performed has to be elaborated in the introduction.
Author reply:Thank you for your suggestion. I have revised the manuscript in the introduction (page4, line3-12).
2.The authors state a significant difference in WTSDS and Tht between the northern and southern population. The author should correct the data of these two parameters for this difference.
Author reply:Thank you for your suggestion. The data of total Tht was wrong. I have corrected the data in the table1 (page14, line8).
3.The authors mention novel AR-mutations. They should not be called mutations, but variants unless the authors show that these new variants are mutations.
Author reply:Thank you for your recommend. I have changed the term mutations to the term variants according your suggestion (page4, line25, page6, line9-12, page16, line12).

Reviewer 2 Report
The study is well done. I can not evaluate the interest of the topic. It is more an analysis of this pathology your country. I consider the study valuable for the clinicians. Good design and pertinent results.
Author Response
Thank you for your comments.
Reviewer 3 Report
Zhao et al. report auxological data from a large population of AIS patients, given the rarity of the situation. The paper confirms previous consolidating data showing that height of the population with AIS is found to be lower than the general male population, but higher than the female one. The reconstruction of the targeted auxological curves was not previously reported. Although evaluating the results we can observe that height, weight and BMI expressed as absolute values ​​or in SDS fall within the range of reference for the age of both the female and male population. The utility of the specific pathology curves could be limited to research at this point.
Although it will be interesting to increase the study population and evaluate the differences between auxological data in patients with the complete vs partial form of AIS, which did not emerge in this study.
The differences in weight and BMI, although they could depend not only on the geographical region of origin, as rightly indicated in the work, but also on genetic predisposition, environmental factors (diet, physical activity). But it might be interesting to discuss the possibility of higher number of AIS patients being overweight or obese associated with impaired effect of sex hormones. Gonadal hormones play an indispensable role in the distribution of lean mass and adipose tissue, as well as body composition as a whole (see Yanase T, Fan W, Kyoya K, Min L, Takayanagi R, Kato S, Nawata H. Androgens and metabolic syndrome: lessons from androgen receptor knock out (ARKO) mice. J Steroid Biochem Mol Biol. 2008 Apr;109(3-5):254-7. doi: 10.1016/j.jsbmb.2008.03.017. Epub 2008 Mar 13. PMID: 18472261.).
Some correction / suggestion:
- I would replace the word "normal" chenese children with chinese pediatric population / general population
- I would omit the sentence on hormonal levels in the introduction
- in the paragraph patients and controls point 2) I would replace with patients with identified pathogenic or ....
- in the paragraph patients and controls point 1) of exlusion criteria I would replace "no" before abnormal
- in the paragraph data collection indicate the formulas used to calculate the bmi and TH
- in the paragraph results why was the division 0-3 yy and 3-16 yy chosen?
- indicate either all as number or transcript (42 and 76 patients)
- i would add 2clinical" before manifestation
- i would replace only micropenis with isolated o hypospadias.
- subdividing the population on the basis of how they grew up (male or female) which criteria could influence the auxological data given that the entire study population was not subjected to the therapies or gonadectomy? In my opinion, this distinction and comparison could be eliminated or argued.
- the bibliography for the second sentence is missing from the discussion
- in the discussion there is no indication of clinical presentations in the CAIS population
- indicate range of hight (167 cm+/- ecc)
- in study limitations replace discussion with not discuss this in the study
Author Response
1. Although it will be interesting to increase the study population and evaluate the differences between auxological data in patients with the complete vs partial form of AIS, which did not emerge in this study.
Author reply:Thank you for your suggetion. I am collecting the more cases of CAIS and PAIS in order to evaluate the difference between auxological data in patients with CAIS and in patients with PAIS. There are not enough dada to draw the growth curve of patients with CAIS and PAIS. I have mentioned it in the paragraph of study limitation. (page18, line16-22)
2. The differences in weight and BMI, although they could depend not only on the geographical region of origin, as rightly indicated in the work, but also on genetic predisposition, environmental factors (diet, physical activity). But it might be interesting to discuss the possibility of higher number of AIS patients being overweight or obese associated with impaired effect of sex hormones. Gonadal hormones play an indispensable role in the distribution of lean mass and adipose tissue, as well as body composition as a whole (see Yanase T, Fan W, Kyoya K, Min L, Takayanagi R, Kato S, Nawata H. Androgens and metabolic syndrome: lessons from androgen receptor knock out (ARKO) mice. J Steroid Biochem Mol Biol. 2008 Apr;109(3-5):254-7. doi: 10.1016/j.jsbmb.2008.03.017. Epub 2008 Mar 13. PMID: 18472261.).
Author reply:Thank you for your suggestion. Your suggestion give me more ideas to discuss the difference of the BMI and weight between children with AIS and general Chinese children. I have revised the discussion according your suggestion (page 16, line 27-page 17, line 3, page 18, line9). I will pay more attention the lean weight and body composition in the future AIS study.
- Some correction / suggestion:
1) I would replace the word "normal" chenese children with chinese pediatric population / general population
Author reply:Thank you for your recommend. I have changed the term “normal children” to “Chinese pediatric population” according to your suggestion (page2, line13, page2, line17, page2, line22, page4, line8, page4, line16, page5, line7, page8, line3-4, page8, line14, page11, line7-8, page12, line7-8, page17, line7, page17, line12-13, page17, line14).
2) I would omit the sentence on hormonal levels in the introduction
Author reply:Thank you for your suggestion. I have revised the manuscript according your suggestion. (page3, line13-page4, line12).
3) in the paragraph patients and controls point 2) I would replace with patients with identified pathogenic or ....
Author reply:Thank you for your recommend. I have changed the sentence according to your suggestion (page4, line24-25).
4) in the paragraph patients and controls point 1) of exlusion criteria I would replace "no" before abnormal
Author reply:Thank you for your recommend. I have changed the sentence according to your suggestion (page5, line1).
5) in the paragraph data collection indicate the formulas used to calculate the bmi and TH
Author reply:Thank you for your recommend. I have added the formulas according to your suggestion (page5, line21-22).
6) in the paragraph results why was the division 0-3 yy and 3-16 yy chosen?
Author reply:The growth curve, influence factors of growth and characters of growth between children aged 0-3 years and aged >3 years show difference. I chose the division 0-3 years and 3-16 years.
7) indicate either all as number or transcript (42 and 76 patients)
Author reply:There were 76 PAIS patients and 42 CAIS patients in the study.
8) i would add 2clinical" before manifestation
Author reply:Thank you for your recommend. I have added the term clinical before manifestation according to your suggestion (page4, line24, page7, line 8-10).
9) i would replace only micropenis with isolated o hypospadias.
Author reply:Thank you for your recommend. I have replaced the terms only micropenis and only hypospadias with the terms isolated micropenis and isolated hypospadias according to your suggestion (page7, line11).
10) subdividing the population on the basis of how they grew up (male or female) which criteria could influence the auxological data given that the entire study population was not subjected to the therapies or gonadectomy? In my opinion, this distinction and comparison could be eliminated or argued.
Author reply:Thank you for your recommend. I have eliminated the distinction and comparison about physical assessment in patients with AIS raised as male and female.
11) the bibliography for the second sentence is missing from the discussion
Author reply:Thank you for your recommend. I have added the bibliography for the second sentence in the discussion. (page16, line8).
12) in the discussion there is no indication of clinical presentations in the CAIS population indicate range of hight (167 cm+/- ecc)
Author reply:Thank you for your recommend. I have added the range of height of CAIS. (page16, line20-21).
13) in the discussion there is no indication of clinical presentations in the CAIS population
Author reply:Thank you for your recommend. I have added the clinical presentations of CAIS in the discussion. (page16, line10-11).
14) in study limitations replace discussion with not discuss this in the study
Author reply:Thank you for your recommend. I have revised the study limitation. (page18, line12-16).

Round 2
Reviewer 3 Report
- English and wording needs to be improved
- line 22 from newborn age..... (abstract)
- line 26 remove and before target height (abstract)
- line 1 replace were with was
- line 5-6 what do you mean male AIS/ female AIS???
- high testosterone level is not the only hormonal feature
- ERalfa : write full name first then abbreviation
- remove the sentence the change of T.....and concerns the whole following paragraph, which is difficult to understand
- use genitalia (in patients and methods) and not genital
- your exlusion criteria were born at the full-term??? so all patients were preterm?
- in results indicate range of median age
- line 12 (resultas ) remove of PAIS
- line 7 in "height for age curve" it is more correct to use the term "length" and not "height" in children under 6 months of age
- the whole following paragraph "height for age curve" is difficult to understand
- figure 3 remove NORMAL chinese boys
- in "physical assesmente" you use the age in years and previously you use 0-36 month, please uniform the data
- discussion line 12 were inherited
- line 18-2 very long sentence
- line 25-27 remove
- line 30 what do you mean for "negative"?
- it is not clear line 31-1
- line 24-26 the timing of puberty in AIS is not clear, please check english
- line 27 you repet closure- closely
- next page line 21 discuss non discussion
- line 28 remove normal
Author Response
English and wording need to be improved
- line 22 from newborn age..... (abstract)
Author reply: Thank you for your suggestion. I have changed the term from age newborn to 11 years to the term from newborn to 11 years of age in the abstract. (page2, line22).
- line 26 remove and before target height (abstract)
Author reply: Thank you for your suggestion. I have revised the sentence in the abstract. (page2, line26).
- line 1 replace were with was
Author reply: Thank you for your suggestion. I have changed the term to the term was in the abstract. (page3, line1).
- line 5-6 what do you mean male AIS/ female AIS???
Author reply: I am so sorry that I have removed the comparison of the patient with AIS raised as male and female in the result. But I forgot to revise the sentences in the abstract. Thank you for your reminder. I have corrected the sentence in the abstract (page3, line4-5).
- high testosterone level is not the only hormonal feature
Author reply: Thank you for your suggestion. The hormonal features of AIS are elevated or normal basal serum testosterone levels associated with high serum LH levels. I have corrected the sentence in the introduction. (page3, line29-page4, line3).
- ERalfa : write full name first then abbreviation
Author reply: Thank you for your suggestion. I have added the full name of Erα, GH and IGF-1 first in the introduction (page4, line9-11).
- remove the sentence the change of T.....and concerns the whole following paragraph, which is difficult to understand
Author reply: Thank you for your suggestion. I have removed the sentence ‘The change of T in patient with AIS may be have an influence on the growth during the childhood’ in the introduction (page4, line12-13). Also, I have revised the sentence according to your suggestion (page4, line12-22).
- use genitalia (in patients and methods) and not genital
Author reply: Thank you for your suggestion. I have changed the term genital to the term genitalia (page4, line4-5, page5, line5, page7, line18, page18, line24).
- your exlusion criteria were born at the full-term??? so all patients were preterm?
Author reply: Thank you for your suggestion. The patients with AIS in the study were born at full-term gestation. It is the inclusion criteria. I am so sorry to make a mistake. I have corrected it in the materials and methods (page5, line9, page4, line13-14).
- in results indicate range of median age
Author reply: Thank you for your suggestion. I have added the range of age in the results (page7, line11).
- line 12 (resultas ) remove of PAIS
Author reply:Thank you for your suggestion. I have removed the term of PAIS in the results according your suggestion (page7, line16).
- line 7 in "height for age curve" it is more correct to use the term "length" and not "height" in children under 6 months of age
Author reply: Thank you for your suggestion. I have corrected the term height to the term length in the results according to your suggestion (page8, line7).
- the whole following paragraph "height for age curve" is difficult to understand
Author reply:Thank you for your suggestion. I have corrected the term height for age curve t to the term height/length-for-age growth curve, the term weight for age curve to the term weight-for-age growth curve and the term BMI for age curve to the term BMI-for-age growth curve in the results according to your suggestion (page8, line3-4, page11, line5-6, page12, line5-6).
- figure 3 remove NORMAL chinese boys
Author reply: Thank you for your suggestion. I have removed the term normal in figure 3 according to your suggestion (page10, line3).
- in "physical assesmente" you use the age in years and previously you use 0-36 months, please uniform the data
Author reply: Thank you for your suggestion. I have corrected the term 0-3 years to the term 0-36 months in the results according to your suggestion (page14, line18, page15, line1-3).
- discussion line 12 were inherited
Author reply: Thank you for your suggestion. I have revised the sentence in the discussion and conclusion according to your suggestion (page16, line12-13).
- line 18-2 very long sentence
Author reply: Thank you for your suggestion. I have revised the sentence in the discussion and conclusion according to your suggestion (page16, line18-19).
- line 25-27 remove
Author reply: Thank you for your suggestion. I have removed the sentence in the discussion and conclusion according to your suggestion (page16, line25-28).
- line 30 what do you mean for "negative"?
Author reply:The paper’ A Androgens and metabolic syndrome: Lessons from androgen receptor knock out (ARKO) mice’ shows the results as follow: Androgen receptor (AR) null male mice revealed late-onset obesity. Male ARKO mice were euphagic compared to the wild-type male controls, but also less dynamic and less oxygen consuming.Transcript profiling indicated that male ARKO mice had lower transcripts for the thermogenetic uncoupling protein 1 (UCP1). We also found enhanced secretion of adiponectin, which is insulin-sensitizing, from adipose tissue in comparison to wild type, which might partly explain why the overall insulin sensitivity of male ARKO mice remained almost intact despite their apparent obesity. In addition, decreased lipolysis rather than increased lipid synthesis was observed, which might also account for the increased adiposity in male ARKO mice. So this paper get the conclusion such as ‘AR plays important roles in male metabolism by affecting the energy balance, and is negative to both adiposity and insulin sensitivity’.
- it is not clear line 31-1
Author reply: Thank you for your suggestion. I have revised the sentence (page16, line30-page 17, line2).
- line 24-26 the timing of puberty in AIS is not clear, please check English
Author reply: Thank you for your suggestion. I have revised the sentence in the discussion and conclusion according to your suggestion (page17, line26-29).
- line 27 you repet closure- closely
Author reply: Thank you for your suggestion. I have revised the sentence in the discussion and conclusion according to your suggestion (page17, line30).
- next page line 21 discuss non discussion
Author reply: Thank you for your suggestion. I have revised the sentence in the discussion and conclusion according to your suggestion (page18, line16-22).
24.line 28 remove normal
Author reply: Thank you for your suggestion. I have removed the term normal in the discussion and conclusion according to your suggestion (page18, line28).